# Effects of Radiant Floor Heating Integrated with Natural Ventilation on Flow and Dispersion in a Newly Decorated Residence

**DOI:** 10.3390/ijerph192416889

**Published:** 2022-12-15

**Authors:** Peng-Yi Cui, Jia-Qi Wang, Feng Yang, Qing-Xia Zhao, Yuan-Dong Huang, Yong Yang, Wen-Quan Tao

**Affiliations:** 1School of Environment and Architecture, University of Shanghai for Science and Technology, Shanghai 200093, China; 2College of Mechanical Engineering, Suzhou University of Science and Technology, Suzhou 215000, China; 3School of Energy and Power Engineering, Xi’an Jiaotong University, Xi’an 710049, China

**Keywords:** numerical simulation, radiant floor heating, natural ventilation, indoor air quality, pollutant dispersion

## Abstract

To date, few studies have been conducted on the characteristics of flow and dispersion caused by indoor radiant floor heating integrated with natural ventilation. In this study, we employed reduced−scale numerical models validated by wind−tunnel experiments to investigate the influence of radiant floor heating integrated with natural ventilation on airflow, heat transfer, and pollutant dispersion within an isolated building. The Richardson number (*Ri*) was specified to characterize the interaction between the inflow inertia force and the buoyancy force caused by radiant floor heating. Several *Ri* cases from 0 to 26.65, coupled with cross− or single−sided ventilation, were considered. Model validation showed that the numerical model coupled with the RNG *k*-*ε* model was able to better predict the indoor buoyant flow and pollutant dispersion. The results showed that the similarity criterion of *Ri* equality should be first satisfied in order to study indoor mixed convection using the reduced−scale model, followed by *Re*−independence. For cross−ventilation, when *Ri* < 5.31, the incoming flow inertia force mainly dominates the indoor flow structure so that the ACH, indoor temperature, and pollutant distributions remain almost constant. When *Ri* > 5.31, the thermal buoyancy force becomes increasingly important, causing significant changes in indoor flow structures. However, for single−sided ventilation, when *Ri* > 5.31 and continues to increase, the buoyancy force mainly dominates the indoor flow structure, causing a significant increase in ACH, thus reducing the indoor average temperature and pollutant accumulation.

## 1. Introduction

In recent decades, economic conditions have continued to improve, and the demands for comfortable and healthy living environments have increased. However, with the increasing demand for building decorations, a large number of new decoration materials, chemicals for daily use, and household appliances have been introduced into residential and public buildings, causing a relative increase in the sources and types of indoor air pollutants, posing a great threat to indoor environmental quality [1].

The main sources of indoor air pollution include interior decoration materials, human metabolism, daily household waste, and outdoor air pollutants [2]. For a newly renovated house, the chemical, physical, biological, and radioactive pollutants emitted by decoration materials have the greatest impact on human health. Good ventilation is the main measure used to improve indoor air quality, especially natural ventilation, which is an economical, effective, and energy-saving green ventilation method [3,4].

Three main methods are usually used to study the characteristics of indoor ventilation and the convective diffusion of pollutants: on-site field measurements, reduced-scale experiments (wind-tunnel or water-channel experiments), and numerical simulations (CFD, computational fluid dynamics). On-site field measurements can be performed to obtain data under real weather conditions; however, due to the huge workload involved, the high cost, and the great influence of rapidly changing external weather conditions, the field measurement method has limited applications and is mostly used in model validations for CFD numerical simulations [5,6,7]. Limited by similarity criteria and the high cost involved, few studies have been conducted on indoor flow, heat transfer, and pollutant diffusion using reduced-scale wind-tunnel experiments alone. Wind-tunnel tests were conducted to study the influence of opening locations and fluctuating wind directions on indoor airflow and pollutant diffusion in cross-ventilated isolated buildings [8,9]; the characteristics of the cross-ventilation flow of a generic block in unsheltered or sheltered conditions [10,11,12]; and the effects of wind directions, window modes, and source positions on cross-unit contamination within a high-rise residential building [13,14,15]. Even so, the amount of data obtained from wind-tunnel tests is still limited; therefore, wind-tunnel tests have been carried out to provide data support for the validation of numerical models.

CFD numerical simulation is an important method to study airflow, heat transfer, and pollutant dispersion within different geometric-scale urban areas that have developed in recent decades. Compared with field measurements and wind-tunnel experiments, CFD numerical simulation has the advantages of low cost, controllable boundary conditions, and abundant data acquisition. For instance, many indoor environment evaluation indexes, such as the age of air, air exchange rate (ACH), pollutant exchange rate (PCH), etc., require sufficient data to be employed [16]. Therefore, numerical simulation provides the possibility for the establishment of various evaluation indexes. However, the accuracy of numerical models is an issue that remains to be solved, and this usually needs to be validated using the experimental data provided by the above two methods. In addition, the numerical simulation method is not limited by the geometric scale of the object studied, and different numerical models can be used to analyze and study the fluid flow and mass transfer processes at the macroscopic, mesoscopic, and microscopic scales [17,18]. For the indoor scale, many previous studies have focused on the impacts of computational settings and parameters on the accuracy of CFD models for cross-ventilated or single-sided ventilated buildings using reduced-scale wind-tunnel measurements (PIV data, tracer gas concentrations, etc.) [19,20,21,22,23]. Subsequently, several case studies were carried out using the validated CFD models to investigate the characteristics of indoor airflow and pollutant dispersion, assessed using various evaluation indexes and with different building physical parameters, such as ventilation modes (cross- or single-sided ventilation), opening dimensions and locations, roof angles, wind directions, and isolated or sheltered ventilated buildings [24,25,26,27,28,29]. Moreover, to conserve calculation resources, Ai and Mak [29] developed reduced-scale CFD models for numerical simulations of wind flow around an isolated building. They found that reduced-scale models using fewer cells than a full-scale model can achieve potentially satisfactory prediction accuracy, although the disadvantage of the limitations of similarity criteria required special attention.

In addition to indoor airflow and pollutant dispersion, the coupling of indoor thermal effects renders the indoor environment complex. Therefore, many previous numerical studies on the validation and verification of models for the influence of thermal effects, coupled with ventilation, on indoor flow structures and pollutant dispersion have been carried out. Gilani et al. [30] conducted model validation and sensitivity analysis, investigating the impacts of the grid resolution, turbulence models, the discretization scheme, and iterative convergence, to evaluate systematically the performance of 3D steady RANS CFD simulations for predicting the temperature stratification within a ventilated room with a heat source located on the center of the ground. In addition to the abovementioned computational parameters, the impacts of other parameters, such as inlet velocity, turbulent kinetic energy, and near wall-treatment, on the performance of CFD simulations of non-isothermal mixing ventilation in an enclosure with a heated floor were evaluated by Kosutova et al. [31]. Moreover, according to the locations of radiant surfaces and air distribution principles, integrated radiant heating/cooling systems that are typically used together with different ventilation modes were reviewed by Zhang et al. [32] to evaluate thermal comfort and indoor air quality for design recommendations. During the last two decades, the increasing use of radiant floor heating applications has gradually replaced the use of traditional radiators and air conditioners, and these have become the main modes of indoor heating for new residential buildings [33,34].

Usually, in a newly decorated and unoccupied residence, heating process can promote the release of toxic and harmful substances such as formaldehyde from building materials, and natural ventilation can effectively and quickly discharge these indoor pollutants. Even when the occupied residences are heated in winter, a fresh air system or reasonable window ventilation is needed to deliver fresh air indoors. Zhou et al. [35] conducted experimental and numerical studies to investigate the effects of ventilation and flow heating systems on the dispersion and deposition of fine particles in an enclosed environment. Their study could be helpful for the design of ventilation and heating systems to remove PM pollution from rooms. To the best knowledge of the authors, studies on the effects of radiant floor heating integrated with natural ventilation on flow and dispersion in a newly decorated room have not been reported to date. One objective of this study was to establish a coupled CFD numerical model for natural ventilation and radiant floor heating effects, validated by means of wind-tunnel experiments. Secondly, the numerical model was used to study the characteristics of flow and pollutant dispersion in a newly decorated room under the coupled effect of radiant floor heating and window ventilation. The results of this study may be helpful in promoting the rapid release and removal of pollutants from building materials in newly decorated rooms through ventilation and heating systems.

## 2. Methodology

### 2.1. Physical Model

The physical model and its dimensions are shown in Figure 1, where the reduced scale ratio is 1:10. Two windows of the same size (*W* × *W*) were included on the windward and leeward walls of a cubic ventilated building (*H* × *H* × *H*). Two natural ventilation patterns were simulated, with cross−ventilation (both windows opening) and single−sided ventilation (window 1 opening, window 2 closed). Moreover, the indoor ground was heated at a constant temperature, *T_f_*, to mimic the radiant floor heating. T_m_ represented the mean air temperature in the room, and T_a_ represented the ambient temperature outside. New furniture (0.2 m × 0.2 m × 0.1 m, full scale) was placed in the indoor ground center, which released pollution gas in the form of VOCs at a constant rate under the heating conditions.

To accurately reproduce the outdoor and indoor flow structures and the heat transfer and pollutant dispersion in the above−described physical model at a reduced scale, it was necessary to follow some similarity criteria. First, the building-height Reynolds number (*Re_H_*) should be satisfied for the outdoor flow and dispersion around the building blocks [36], which can be specified as follows:
(1)$${Re}_{H}=U_{H}H/\nu$$


Meanwhile, the window Reynolds number (*Re_w_*) of the room can be defined to characterize the indoor flow structure and pollutant dispersion, defined by:
(2)$${Re}_{w}=U_{w}H/\nu$$

In addition, to mimic the interaction of the buoyancy force caused by the radiation heating floor and the freestream inertial force from the natural ventilation, the Richardson number (*Ri*, the ratio of Grashof number (*Gr*) to the square of *Re_w_*) is adopted and defined as follows:
(3)$$Ri=\frac{Gr}{Re_{w}{}^{2}}=\frac{ga_{v}(T_{f}-T_{a})H}{U_{w}^{2}}$$
where *U_H_* is the freestream wind speed at the building height *H* (m/s); *U_w_* is the mean wind velocity through the open window (m/s); ν is the kinematic viscosity of air; *a_v_* is the thermal expansion coefficient; and *g* is the gravitational acceleration. In this study, seven *Ri* values (from 0 to 26.65) for reduced-scale physical models were investigated, as shown in Table 1, with *U_H_* changing from 0.1 to 6 m/s, and the relative parameters *Re_H_*, *U_w_*_,_ *Re_w_*, and *Gr* are also listed.

### 2.2. Numerical Method

#### 2.2.1. Governing Equations

The governing equations used to describe the indoor airflow, heat transfer, and pollutant dispersion, considering the thermal effect from the radiation floor, are shown as follows, including the continuity equation in incompressible form,
(4)$$\frac{\partial u_{j}}{\partial x_{j}}=0$$
as well as the RANS (Reynolds-averaged Navier–Stokes) equation,
(5)$$\frac{\partial (\rho u_{j}u_{i})}{\partial x_{j}}=-\frac{\partial p}{\partial x_{i}}+\frac{\partial }{\partial x_{j}}\left(\left(\mu +\mu_{t}\right)\left(\frac{\partial ui}{\partial x_{j}}+\frac{\partial uj}{\partial x_{j}}\right)\right)+\rho g_{i}$$
the energy equation,
(6)$$\frac{\partial \left(\rho u_{j}T\right)}{\partial x_{j}}=\frac{\partial }{\partial x_{j}}\left(\left(\frac{\mu }{\mathrm{Pr}}+\frac{\mu_{t}}{\sigma_{t}}\right)\frac{\partial T}{\partial x_{j}}\right){+S}_{T}$$
and the species transport equation for passive advection–diffusion,
(7)$$\frac{\partial (\rho u_{j}c_{i})}{\partial x_{j}}=\frac{\partial }{\partial x_{j}}\left((\rho D_{i,m}+\frac{\mu_{t}}{Sc_{t}})\frac{\partial ci}{\partial x_{j}}\right)+S_{c}$$
where *u_i_* is the time-averaged velocity to be solved (m/s); *u**_j_*** is the air velocity vector (m/s); ρ is the air density (kg/m^3^); p is the pressure (Pa); *μ* and *μ_t_* are the molecular viscosity and turbulent viscosity, respectively (Pa·s); *T* is the temperature (K); Pr is the Prandtl number; σ***_t_*** is the turbulent Prandtl number (also written as *Pr_t_* = 0.85); S*_T_* is the heat source term (W/(m^3^·s)); *c**_i_*** is the mass concentration of pollutants (kg/m^3^); *D**_i,m_*** is the molecular diffusion coefficient for pollutants (m^2^/s); *S_c_* denotes the pollution source term (kg/(m^3^·s)); and *Sc_t_* is the turbulent Schmidt number (=0.7) [37].

The discrete ordinates (DO) radiation model was employed to consider the radiation heat transfer between the heated radiant floor (aluminum) and the building’s inner walls (wood). The DO model solves the radiative transfer equation (RTE) for a finite number of discrete solid angles, each associated with a vector direction $\vec{s}$ which is fixed in the global Cartesian system (*x*, *y*, *z*). Herein, the air in the ventilated room model is treated as a non-participating medium due to its optical thickness being almost zero. The solid surfaces were assumed to be diffusive and gray, and the values of their internal emissivity were selected as follows: 0.9 and 0.89 for radiator and walls (wood), respectively [38].

The model validation, as described in Section 4.1, proved that the RNG *k*-*ε* model was able to better predict the characteristics of the indoor airflow, heat transfer, and pollutant diffusion. The RNG *k*-*ε* turbulence model includes the *k* (turbulent kinetic energy) and *ε* (the dissipation rate of *k*) equations proposed by Yakhot and Orszag [39].

#### 2.2.2. Computational Domain and Boundary Conditions

Figure 2 shows the computational domain and boundary conditions. According to Tomanaga et al. [40], the independence of the computational domain was implemented, indicating that the appropriate computation domain was finally determined as *Lx* × *Ly* × *Lz* = 25*H* × 13*H* × 7*H*. Here, the inlet, top, right, and left planes were all 6*H* away from the corresponding building model surfaces. Furthermore, the outlet plane was 18*H* away from the leeward surface of the building model to ensure the occurrence of fully developed flow there. The computational domain was discretized using ANSYS ICEM 18.0 by means of structured grids with variable sizes: the finest grid resolution with a cell magnitude of 0.02–0.05*H* inside the indoor scale; a coarser resolution close to the building and ground surfaces with a magnitude of 0.05–0.1*H*; and a gradually coarse mesh from the building and ground surfaces to the domain boundaries, using an increasing ratio of 1.12–1.16.

Three grid resolution cases are shown in Figure 3a–c, where the coarse, fine, and finer grid numbers are 1.80 million, 5.80 million, and 8.80 million, respectively. Figure 3d indicates that the total independent grid number was 5.8 million. Table 2 summarizes the boundary conditions for the numerical models.

## 3. Wind-Tunnel Experimental Setup

The wind-tunnel experiments for numerical validation were carried out in the Environmental Wind Tunnel Laboratory at University of Shanghai for Science and Technology (EWT Lab at USST), which is an open-circle atmospheric boundary layer wind tunnel with test section dimensions of 2.5 m in width, 1.8 m in height, and 25 m in length. The wind speeds on the inlet plane can be adjusted continuously from 0.1 to 20 m/s, and the scaled profiles of the atmospheric boundary layer (ABL) flow can be generated by means of six Irwin-type spires and roughness elements of different sizes, arranged regularly within the 15 m long development section (Figure 4). The building model, made of wood, with a reduced scale ratio of 1:10 was placed at the center of a circular turntable with a diameter of 2 m in the test section.

During the wind-tunnel tests, the mean velocity and turbulence intensity were measured using a TSI IFA 300 3D constant-temperature hot-film anemometer with a high accuracy of ±0.1%. Aluminum heating panels with adjustable temperatures from ambient temperature to 400 °C were placed on the indoor ground to simulate the thermal convection caused by radiant floor heating. The pollution source was placed on the indoor ground center and was connected to the tracer gas release system through the bottom pipes. After releasing the gas, the sampling pumps (with a sampling rate of 150 mL/min) could collect the tracer gas into sampling bags by connecting the sampling tubes (D = 2 mm) at different heights (Figure 4b). The tracer gas concentrations in the sampling bags were measured using a DB−600T SF_6_ quantitative detector with a measurement accuracy of ±1 ppm. The normalized concentration *K* is specified as follows: (8)$$K=\frac{C_{r}U_{H}H^{2}}{C_{e}Q_{e}}$$
where *C**_r_*** and *C**_e_*** are the volume fractions of the measured SF_6_ (ppm) and the pollution source (=2.5 × 10^5^ ppm). T-type thermocouples were applied to measure the air temperature along the indoor vertical centerline when the indoor ground was heated. The dimensionless temperature *θ* is specified as: (9)$$\theta =\frac{T-T_{a}}{T_{f}-T_{a}}$$
where *T* is the measured air temperature value (K). In addition, the laser sheet-CCD concentration testing system was also used to characterize the pollutant dispersion in the room under different *Ri* cases, which allowed the high-speed camera to capture the diffusion features of trace smoke displayed on a sheet laser.

## 4. Results and Discussion

### 4.1. Validation of Numerical Models

The numerical model validations were conducted by comparing the wind-tunnel experimental data (WT data) for cross-ventilation under *Ri* = 0.27 with the established numerical models by adopting three high—*Re* turbulence models, namely, standard, RNG, and realizable *k*-*ε* (SKE, RNG, RLKE) turbulence models. First, Figure 5a shows a comparison of normalized velocity (*U*/*U*_0_) distributions observed along the indoor vertical centerline. It can be noted that the RNG model performed better when predicting the indoor flow structure than the other two turbulence models, although the results from the three turbulence models showed similar trends. Wind speed tended to be higher in the middle and lower at both ends. A comparison of the dimensionless concentrations (*K*) along the indoor vertical centerline is depicted in Figure 5b, with the pollutant concentration decreasing sharply with increasing height. The same trends can also be observed in Figure 5c, where the dimensionless temperature (*θ*) distributions between WT tests and numerical models are depicted. Based on the above comparisons, we found that the distributions of *U*/*U_0_*, *K*, and *θ* in the room obtained using the RNG model agreed better with the WT data (with mean deviations of approximately less than 6%), followed by the RLKE model, and the SKE model performed the worst.

For further confirmation of the model validation, the characteristics of flow and dispersion, measured using a laser sheet—CCD concentration testing system under different *Ri* cross-ventilation conditions, have been depicted in Figure 6a–f. Here, the red lines drawn from the visualization of smoke plumes represent the streamlines, and the smoke diffusion can be used to characterize the pollutant dispersion under different *Ri* conditions. By comparing these results with the following streamlines and concentration distributions under the same *Ri* conditions (*Ri* values ranging from 0 to 11.11), we can further confirm that the RNG model showed high numerical accuracy in predicting the indoor flow and dispersion under the joint action of different inertia forces and thermal buoyancy forces.

### 4.2. Flow and Dispersion Characteristics under the Same Ri but Different Re Values

Figure 7(a1–a3) depict the flow patterns and pollutant dispersion under three cases with the same *Ri* value (=0.27) but different *Re* and *Gr* values. It is worth noting that their flow patterns and pollutant distributions can be regarded as almost the same from the point of view of the numerical simulation. That is because, considering forced convection by cross-ventilation, the flow structures in the room entered the region of Re-independence (three values of Re***_w_*** greater than 15,000, Cui et al. [41]), which means that the flow structures will change little with increasing *Re*. In such situations, the *Ri* value becomes the only influencing factor. Therefore, the concept of *Re*-independence is not only valid for pure forced convection flow but is also applicable to mixed convection involving thermal buoyancy flow. However, Figure 5b shows the distributions of *K* against the indoor vertical centerline under the above three cases, which indicates that the relative deviation became much smaller with increasing *Re* values, although this deviation among the three cases was already less than 5%. Based on the above analysis, we can conclude that the similarity criterion of *Ri* equality should be first satisfied by using the reduced−scale model to study the indoor mixed convection and should then be followed by *Re*-independence.

### 4.3. Characteristics of Indoor Flow Structures with Varying Ri Values

#### 4.3.1. Contours of U/U_0_ against Increasing *Ri*

Figure 8a–g show the contours of dimensionless velocity (*U/U*_0_) plotted against increasing *Ri* numbers for cross−ventilation cases. With increasing *Ri* values, ranging from 0 (pure natural ventilation) to 26.65, several features can be observed. Firstly, when *Ri* < 5.31, with the increasing *Ri*, the indoor flow distribution changed little (similar flow structure), which shows that the air entered the room from the windward side window and flowed out from the leeward side window horizontally, forming a clockwise vortex at the lower part of the room. This is because when *Ri* is small (less than 5.31), the inertial force of the incoming flow plays a leading role in dominating the structures of the indoor flow field; meanwhile, the thermal buoyancy force caused by the radiant floor heating is small and can be ignored. Secondly, when *Ri* is greater than 5.31, we can observe that the indoor flow fields changed significantly, showing that the airflow horizontally entered the room from the leeward window and was deflected downward, squeezing the vortex in the lower part into the left corner of the room. As the *Ri* continued to increase, the downward deflection of the airflow entering the room became more obvious, and the vortex was gradually squeezed to become smaller and smaller. This is because when the *Ri* value was greater than 5.31, the buoyance force caused by the radiant heating floor could no longer be ignored, and it played an increasingly important role in changing the indoor flow structure under the combined action of the inflow inertial force and the increase in *Ri*. In addition, with the increase in *Ri*, the gradually enhanced thermal buoyant force caused more and more thermal airflow to gather in the top of the room, thus pushing down the airflow entering the room from the windward window in an increasingly obvious manner. Then, the two streams are mixed together and flow out of the room through the leeward window. The characteristics of the flow field discussed above are highly consistent with the flow field visualization results (the red streamlines) obtained from the WT test, shown in Figure 6, which again verifies the reliability of the numerical model adopted in this study.

The contour plots of dimensionless velocity (*U/U*_0_) on the vertical center plane against increasing *Ri* values for single-sided ventilation cases are depicted in Figure 9. The biggest difference from the cross-ventilation results was the fact that there was less airflow entering the room in the single-sided ventilation cases. However, with the increase in *Ri*, the following characteristics of indoor flow structures can be obtained. Firstly, when *Ri* = 0, the incoming flow was blocked by the indoor positive air pressure and was then deflected downward and entered the room vertically along the lower edge of the window. Then, two vortices were formed in the right part of the room: a clockwise one at the bottom, and another counterclockwise one at the top. However, these flow structures were very unstable, because when the indoor floor started to heat up (with *Ri* equal to 0.04, as shown in Figure 9b), the two small vortices merged to form a large counterclockwise vortex. Secondly, when the *Ri* value was less than 5.31, as the *Ri* increased, we were able to note that the large vortex maintained a relatively stable structure; however, the mean velocities in the room increased gradually. When the incoming flow decreased, the mean indoor velocity increased, indicating that the thermal buoyancy force caused by radiant floor heating gradually dominated the indoor flow structures. Thirdly, as the *Ri* continued to increase (Figure 9f,g), the large indoor counterclockwise vortex was gradually compressed by the thermal airflow at the top to right−bottom of the room. Furthermore, the windward window was obviously divided between the upper and lower parts, with the external airflow entering the room from the lower part of the window and forming the main vortex, combined with the indoor thermal airflow.

#### 4.3.2. Mean ACH Plotted against the Square *Ri*

To quantitatively evaluate the ventilation capacity as the *Ri* value increased, we made use of the air exchange rate (*ACH*) across the open window plane, i.e., a method which has been adopted in several studies [16,42], consisting of the mean ($\bar{\mathrm{ACH}}$) and turbulence (${\mathrm{ACH}}^{\prime }$) components as follows: (10)$$\mathrm{ACH}=\bar{\mathrm{ACH}}+{\mathrm{ACH}}^{\prime }\;$$
where $\bar{\mathrm{ACH}}$ is induced by the exchange of the mean velocity and ${\mathrm{ACH}}^{\prime }$ is induced by the fluctuation velocity. According to the approach proposed by Li et al. [42], the *ACH*′ across the open window plane can be determined based on the turbulent kinetic energy *k* as follows: (11)$${\mathrm{ACH}}_{W_{1}}^{\prime }+=-{\mathrm{ACH}}_{W_{1}}^{\prime }-=\frac{1}{\sqrt{6}}\int_{\Gamma_{1}}\sqrt{k}d\Gamma_{1}\;$$
(12)$${\mathrm{ACH}}_{W_{2}}^{\prime }+=-{\mathrm{ACH}}_{W_{2}}^{\prime }-=\frac{1}{\sqrt{6}}\int_{\Gamma_{2}}\sqrt{k}d\Gamma_{2}\;$$
where *Γ*_1_ = *Γ*_2_ = *W* × *W* is the area of the open window, *W***_1_** and *W*_2_. The *ACH* value can be normalized based on the room volume *V* (= *H* × *H* × *H*) and the reference time *T* (= *H*/*U**_H_***). Table 3 and Table 4 illustrate the dimensionless *ACH*+/− for cross-/single-sided ventilation cases under different *Ri* values, respectively.

In Table 3, we can first note that with the increase in *Ri*, the dimensionless $\bar{\mathrm{ACH}}$− becomes larger and larger, compared to the dimensionless $\bar{\mathrm{ACH}}$+, with the maximum deviation reaching 20% when *Ri* = 26.65. This is because with the increase in *Ri*, indoor air is heated by the radiation floor and the degree of thermal expansion increases, causing the average momentum of the airflow to flow out from the leeward window, increasing more and more significantly than that entering the room from the windward window. Second, although the dimensionless $\bar{\mathrm{ACH}}$ across *W*_1_ and *W*_2_ changes with the increase in *Ri*, the dimensionless *ACH*′ on these two window planes is almost identical (*ACH’**_W_***_1_+ = *ACH’**_W_**_2_*−). Therefore, the dimensionless *ACH*+ also differs from the dimensionless *ACH*− (*ACH*− > *ACH*+) more and more significantly with the increase in *Ri*, which is different from the numerical results obtained for *ACH*+ = *ACH*− under the isothermal cases reported in some previous studies [16,42,43]. Third, when *Ri* is less than 5.31, the proportion of the dimensional $\bar{\mathrm{ACH}}$ is about 70%, which is about 30% of the value of *ACH*′. However, as the *Ri* value continues to increase, the proportion of the dimensional $\bar{\mathrm{ACH}}$ gradually decreases slightly, which is because, under the studied cross-ventilation cases, the rate of the contribution of the dimensional $\bar{\mathrm{ACH}}$ is always larger than that of *ACH*′, and the increase in *Ri* leads to an insignificant increase in the dimensional $\bar{\mathrm{ACH}}$ across *W*_2_ ($\bar{\mathrm{ACH}}$−). In Table 4, for the single-sided ventilation cases, we can observe that the largest difference was the fact that as the *Ri* value increased, the proportion of the dimension *ACH*′ increased significantly from 75 to 90% (with *Ri* values from 0 to 1.86), then decreased from 90% to about 26% (with *Ri* values from 1.86 to 26.65). This indicates that the air exchange efficiency between indoor and outdoor air was mainly dominated by turbulence velocity when *Ri* was less than 5.31; when *Ri* = 26.65, it was mainly dominated by mean velocity; and when *Ri* = 11.11, it was dominated by both turbulence and mean velocities.

To further confirm the effects of wind and thermal buoyancy on the indoor and outdoor air exchange, Figure 10 depicts the mean dimension *ACH* (the average value of *ACH*+ and *ACH*−) against the square *Ri* value for cross- and single-sided ventilation cases. Based on the variation trend shown in Figure 10, several features can be noted. Firstly, the dimensionless *ACH* values for cross-ventilation were about five times greater than those for single-sided ventilation under the same *Ri* cases, indicating that cross-ventilation was more conducive to indoor ventilation capacity. Secondly, for cross-ventilation, when *Ri* was no more than 5.31, it seems that the dimensional *ACH* (about 0.15) was independent of the *Ri* numbers, which indicates that under these cases, the air exchange between indoors and outdoors was mainly dominated and driven by the wind, whereas thermal buoyancy caused by the radiation floor heating could be ignored. However, when *Ri* was greater than 5.31, as the *Ri* value increased, the dimensionless *ACH* decreased slightly, indicating that the effect of thermal buoyancy could not be ignored, and the indoor flow structures and ventilation capacity were dominated by the combined action of wind and thermal buoyancy. Thirdly, for the single-sided ventilation, when *Ri* was no more than 0.27, the dimensionless *ACH* was independent of the *Ri* value and the thermal buoyancy could be ignored; when *Ri* was between 0.27 and 5.31, as the *Ri* increased, the dimensionless *ACH* decreased slightly, with wind and buoyancy working together to reduce the indoor ventilation capacity. However, as *Ri* continued to increase, although the incoming wind speed (*U**_w_***) also continued to decrease, the dimensionless *ACH* increased significantly, which indicates that the thermal buoyancy mainly dominated the indoor flow structures and air exchange capacity with the outdoors. Through the above analysis, we can conclude that the thermal buoyancy effects caused by radiant floor heating will dominate the indoor flow more quickly in the case of single-sided ventilation with increasing *Ri* values.

### 4.4. Characteristics of Indoor Temperature Distributions with Varying Ri

#### 4.4.1. Contours of *θ* against Increasing *Ri*

Figure 11 depicts the contours of dimensionless temperature (*θ*) against increasing *Ri* values in the vertical center planes for cross-ventilation cases. The following features can be observed. First, in Figure 11, it can be noted that when *Ri* < 5.31 (Figure 11a–d), the indoor temperature distributions could be divided into three parts based on the incoming wind: the upper part, the middle part, and the lower part. Here, the lower part was the hottest since it was closest to the heat source, followed by the upper part, and the temperature of the middle part was the lowest due to the incoming flow of fresh air. With the increase in *Ri*, the indoor mean temperature increased, with an area of high temperature in the bottom left corner of the lower part increasing significantly, and an area of low temperature in the middle part also greatly decreasing. This is because when *Ri* was less than 5.31, the incoming wind still dominated the indoor flow and dispersion. With the increase in *Ri*, although the indoor ventilation capacity remained basically unchanged (*ACH* remained unchanged in Figure 10), the increasing heat release from the radiation floor mainly caused the indoor temperature to rise gradually. Second, when *Ri* was greater than 5.31, as *Ri* increased, the incoming wind and thermal buoyancy together dominated the indoor flow structure, causing the temperature of the upper part and the right-middle part to rise significantly. This indicates that as the *Ri* continued to increase, the incoming wind was bent downward to the right corner (Figure 8f,g) to carry more heat to the upper part. Moreover, as the *Ri* value increased, the decreasing *ACH* coupled with the increasing heat release from the radiation floor together led to a significant increase in temperature, especially in the upper part of the room.

In Figure 12, for the investigation of single-sided ventilation, we can note that the high temperature occupied the greatest part of the room, and only a small part near and below the window, where the airflow entered the room, showed relatively lower temperature distributions. When *Ri* was less than 5.31, the mean temperature in the room increased significantly with the increase in *Ri*. However, when *Ri* was greater than 5.31 with a continuous increase, the indoor mean temperature decreased, and there was a greater heat flow out of the room from the upper window, climbing along the windward wall to the roof. This is because when *Ri* was less than 5.31, although the indoor ventilation capacity (*ACH*) remained almost unchanged with the increase in *Ri*, the release of heat from the radiation floor continued to increase, causing the indoor temperature to rise gradually. However, when *Ri* was greater than 5.31 for the single−sided ventilation model, the indoor ventilation capacity increased significantly with the continuous increase in *Ri*, and although the heat release rate also increased, it was always smaller than the amount carried out by the airflow. Therefore, based on this analysis, we can conclude that the indoor temperature distributions depended on the ventilation mode and the *Ri* value. For the studied *Ri* numbers, the mean temperature in the room increased with increasing *Ri* in the cross−ventilation cases; however, in the single−sided ventilation model, as the *Ri* increased, the mean indoor temperature increased first and then decreased.

#### 4.4.2. Mean h against the Square *Ri*

To further evaluate the effect of ventilation capacity and the *Ri* value on the indoor temperature distributions, Figure 13 depicts the mean indoor surface convective heat transfer coefficient *h* against the square *Ri* for cross- and single−sided ventilation cases. Here, *h* is specified as follows: (13)$$h=\frac{q}{T_{f}-T_{m}}$$
where *q* is surface heat flux (W/m^2^), and *T**_m_*** is the mean temperature in the room (K). In Figure 13, it can first be noted that for the cross-ventilation cases, when *Ri* was less than 5.31, *h* decreased sharply from 35 to 20 W/(m^2^·K) with increasing *Ri*, indicating that the mean indoor *h* depended strongly on *Ri*. This is because when *Ri* < 5.31, the inertial force of incoming wind dominated the indoor flow structure, and as the *Ri* increased, the weakened ventilation capacity led to the accumulation of heat and an increase in the mean temperature in the room. Together with the increase in the temperature difference between the heating floor and indoor air, this resulted in a significant decrease in the mean *h*. However, when *Ri* was greater than 5.31, the *h* curve was quite mild, meaning that *h* had almost nothing to do with *Ri*, which was because, although the temperature difference between the heating floor and the indoor ambient air increased with the continuous increase in *Ri*, the ventilation capacity also increased due to the effect of enhanced thermal buoyancy. Second, for the single-sided ventilation cases, with the increase in *Ri*, the mean values of *h* first decreased sharply (when *Ri* was no more than 1.86), and then increased mildly when *Ri* was between 1.86 and 11.11 and increased greatly when *Ri* was greater than 11.11. Unlike the cross-ventilation case, the main reason for the increase in the mean *h* with the continuous increase in *Ri* for the single-sided ventilation cases is that when *Ri* was greater than 1.86, the thermal buoyancy force caused by the heating floor could no longer be ignored, especially for *Ri* values greater than 11.11, the indoor flow structure was mainly dominated by thermal buoyancy, and the ventilation capacity increased sharply. Third, based on the above analysis, we can conclude that the mean *h* in the room was mainly dependent on the relative strength of the incoming inertial wind force and the thermal buoyancy force. When the incoming wind dominated the indoor flow structure, *h* decreased significantly with the increase in the *Ri* value, and when the indoor flow structure was dominated by both incoming wind and thermal buoyancy, *h* depended little on the *Ri* value (which was unchanged with the increase in the *Ri* value); however, when the thermal buoyancy force dominated the indoor flow structure, *h* increased significantly with the increase in *Ri*.

### 4.5. Characteristics of Indoor Pollutant Dispersion with Varying Ri Values

#### 4.5.1. Contours of K against Increasing *Ri*

To reveal the effects of different *Ri* values and ventilation modes on the indoor pollutant distribution, Figure 14 shows the contours of the dimensionless concentration (K) in the indoor vertical center plane, plotted against varying *Ri* values for cross ventilation cases. We can observe that the distribution of K against the increasing *Ri* values are similar to the dimensionless temperature distributions shown in Figure 11. Firstly, when *Ri* was less than 5.31, the indoor pollutant distributions could also be divided into three parts based on the incoming wind: the upper part, middle part, and lower part. Here, the lower part was more seriously polluted due to the proximity of the pollutant source, followed by the upper part, and then the concentration in the middle part was the lowest due to the incoming flow of fresh air. Secondly, it can be noted that when *Ri* < 5.31, the indoor pollutant distributions in the three parts changed little with the increase in *Ri*, with the high pollutant concentration in the lower part moving from the bottom center to the bottom left corner and the pollutant concentration in the middle part increasing slightly with the increase in *Ri*. This is because when *Ri* was less than 5.31, the inertial force of the horizontal incoming wind still dominated the indoor flow and the pollutant dispersion and the buoyancy force could be ignored, resulting in the formation of the above three parts. However, as shown in Figure 8, as the buoyancy force (*Ri*) increased, the clockwise vortex in the lower part moved gradually from the bottom right (*Ri* = 0) to the bottom center (*Ri* = 0.04, 0.27, and 1.86), causing the distribution to show a high pollutant concentration. Thirdly, when *Ri* was greater than 5.31, the thermal buoyancy force caused by the radiation floor could no longer be ignored, but rather acted together with the inertial force of the incoming flow to cause the incoming flow to be deflected from the horizontal direction to the downward-right area. Therefore, as the *Ri* value increased, the high−concentration area of pollutants in the lower part was gradually deflected from the left corner to the right corner and diffused out through the leeward window.

Figure 15 depicts the contours of K, plotted against increasing *Ri* values for the single-sided ventilation cases. Unlike the cross−ventilation results shown in Figure 14, the indoor pollutant distributions for single−sided ventilation changed greatly with increasing *Ri* values. First, when *Ri* = 0 (no thermal effect), because it was difficult for the fresh airflow from the outside to enter the room, there was no stable flow field structure formed by the thermal convection in the room, so the pollutants were evenly distributed in the room. Second, when *Ri* > 0, a stable counterclockwise main vortex was formed in the room under the influence of the thermal buoyancy caused by the heating floor coupled with the incoming wind (Figure 9). Therefore, under the action of this counterclockwise vortex, the indoor high−concentration pollutants mainly gathered in the right and top areas of the room. With an increase in the *Ri* value from 0.04 to 5.31, the values of the dimensionless *ACH* of the single−sided ventilation cases remained almost constant or even decreased slightly (Figure 10). However, the *K* distributions in the room seemed to decrease gradually, particularly in the area of high concentration at the right and top of the room, which decreased significantly. This is because with the increase in *Ri*, the mean dimensionless velocities in the room increased gradually (see Figure 9) due to the increasingly important role of the thermal buoyancy caused by the heating ground. Therefore, in the single-sided ventilated room, the effect of thermal buoyancy on indoor air flow and pollutant dispersion was more obvious than that in the cross−ventilated room, because the inertial force of outdoor flow had less of an influence on the flow structure in the single−side ventilated rooms. Third, as the *Ri* continued to increase (*Ri* from 11.11 to 26.65), the average values of K in the room continued to decrease and the characteristics of the K distribution also changed slightly, showing that the high concentration gradually moved from the top to the center of the room. This is mainly because when *Ri* value was greater than 11.11, the thermal buoyancy force dominated the indoor flow structures, and more and more thermal airflows gathered at the top of the room, squeezing the interior counterclockwise vortex down to the lower part (Figure 9f,g). Furthermore, there were obvious thermal airflows out of the room through the upper part of the window, taking away more indoor air pollutants. In addition, unlike the cases of *Ri* values lower than 11.11, due to the strong thermal buoyancy, the air pollutants flowing out of the window climbed up along the vertical façade of the building, then skipped over the roof and traveled down to the wind.

#### 4.5.2. Mean *K* in the Room against the Square *Ri*

To further analyze the indoor concentrations quantitatively, Figure 16 depicts the mean K in the room against the square *Ri* for cross− and single−sided ventilation cases. Based on the curves, we can first note that the mean concentrations in the room for cross-ventilation cases were much smaller than those observed for the single-sided ventilation cases; however, this difference decreased with the increase in the *Ri* value. Second, we can observe that with the increasing *Ri*, the mean concentrations in the room for the cross−ventilation cases remained almost unchanged (less than 25) against the increasing *Ri*; however, the mean values of K for the single−sided ventilation cases sharply decreased from 230 to about 50, which indicates that the indoor pollutant dispersion for the single−sided ventilation cases depended strongly on the *Ri* value. This is because it was difficult for external airflows to enter the single−sided ventilation room, and the thermal buoyancy force caused by the radiation heating floor rapidly dominated the indoor flow structure and pollutant dispersion. In addition, by comparing Figure 10 and Figure 16, we can note that the dimensionless *ACH* did not seem to correlate well with the mean concentration in the room, which is because the diffusion of indoor pollutants to the outdoors was also determined by the indoor flow structure and location of the pollution source. For instance, in the cross-ventilation cases, when *Ri* was less than 5.31, the inertial force dominated the indoor flow field, and the indoor flow structures and the dimensionless *ACH* remained almost the same (Figure 8a–d and Figure 10); therefore, the mean K in the room also remained constant. However, as the *Ri* continued to increase, although the dimensionless *ACH* decreased from 0.15 to 0.13 (Figure 10), the mean K in the room also decreased slightly (Figure 16). This is because when *Ri* exceeded 5.31, although the *ACH* decreased, the indoor flow structure changed greatly (with the airflow blowing directly towards the pollution source), resulting in an increase in the effective ventilation rate that could carry more pollutants to the outside. To sum up, it can be noted that the mean indoor pollutant concentration was not only related to the dimensionless *ACH* but was also closely related to the indoor flow structure, i.e., the effective ventilation rate must be increased to reduce the indoor pollutant accumulation.

## 5. Conclusions

In this study, reduced-scale numerical models were used to investigate the influence of radiant floor heating integrated with natural ventilation on indoor flow and dispersion. The numerical model was validated by means of wind−tunnel tests. Seven *Ri* values ranging from 0 to 26.65, coupled with cross- or single−sided ventilation, were considered. Our conclusions are as follows: (1)It can be noted that the similarity criterion of *Ri* equality should be satisfied first by using a reduced−scale model to study the indoor mixed convection, followed by Re−independence, and the *Re* value should be as large as possible in order to reduce the deviation caused by the reduced scale.(2)Indoor flow structures:

For cross-ventilation, when *Ri* was less than 5.31, the indoor flow structures remained almost the same, and the inertial force dominated the indoor flow field. When *Ri* was greater than 5.31, the dimensionless *ACH* vs. *Ri*^1/2^ results confirmed that the indoor flow field was dominated by the wind and buoyancy force together. However, for single−sided ventilation cases, when *Ri* was greater than 5.31, the buoyancy force dominated the indoor flow structure, and the incoming inertial force could be ignored. Meanwhile, the dimensionless *ACH* increased significantly. Therefore, we found that the thermal buoyancy effects caused by radiant floor heating dominated the indoor flow structure more easily in single-sided ventilation with increasing *Ri*.

(1)Indoor temperature distributions:

In the case of cross−ventilation, the indoor temperature distributions could be divided into three parts according to the incoming wind, and the high temperatures were mainly concentrated in the lower part. With increasing *Ri*, the indoor mean *h* decreased sharply first, then remained almost unchanged (equal to about 20) when *Ri* was greater than 5.31. For single−sided ventilation, high temperatures occupied most of the room. Moreover, as the *Ri* value increased, the indoor mean *h* for the single-sided ventilation scenario first decreased sharply, then increased mildly, and finally increased significantly when *Ri* was greater than 11.11.

(2)Indoor pollutant dispersion:

In the case of cross−ventilation, when *Ri* was less than 5.31, the indoor pollutant distribution in the three parts changed little with the increase in *Ri*, and the lower part was seriously polluted due to the proximity of the pollutant source, followed by the upper part. When *Ri* was greater than 5.31, as *Ri* continued to increase, the high concentration of pollutants in the lower part was gradually deflected from the left corner to the right corner. However, for the single-sided ventilation scenario, the high concentration of pollutants was mainly concentrated in the bottom right corner and along the right wall, and the mean *K* in the room was greatly reduced with the increase in *Ri*.

In addition, we found that the indoor pollutant concentration was not only related to the dimensionless *ACH* but was also closely related to the indoor flow structure, i.e., the effective ventilation rate must be increased to reduce the indoor pollutant concentration and accumulation.

Limitations: The scenario simulated in this study was that in a newly decorated and unoccupied residence, in which the heating system could promote the release of VOCs such as formaldehyde from the building materials, natural ventilation from windows could quickly remove the pollutants from the room, so there was no discussion of thermal comfort. In future works, we will continue to investigate the coupled effects of indoor heating and fresh air systems on indoor pollutant removal and thermal comfort in occupied residences.

## Figures and Tables

**Figure 1 ijerph-19-16889-f001:**
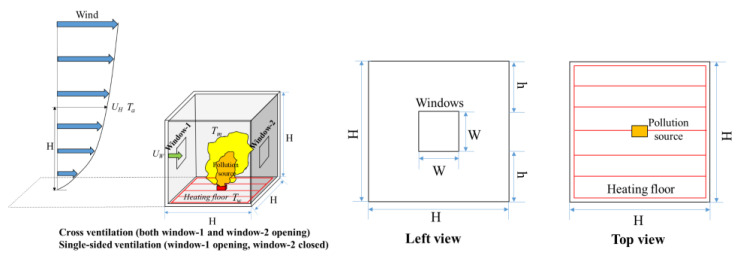
Physical model.

**Figure 2 ijerph-19-16889-f002:**
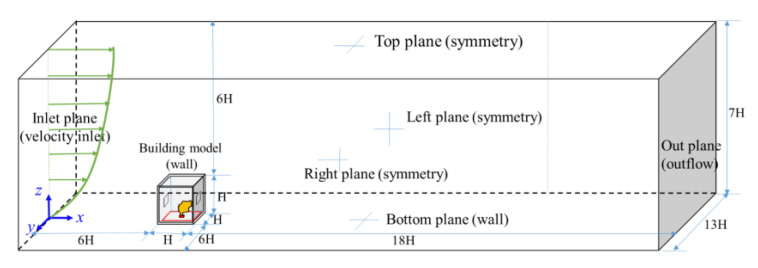
Computational domain and boundary conditions.

**Figure 3 ijerph-19-16889-f003:**
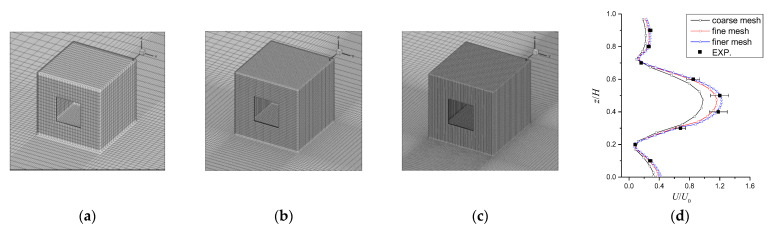
Mesh generation and grid independence analysis. (**a**) coarse mesh; (**b**) fine mesh; (**c**) finer mesh and (**d**) grid independence analysis.

**Figure 4 ijerph-19-16889-f004:**
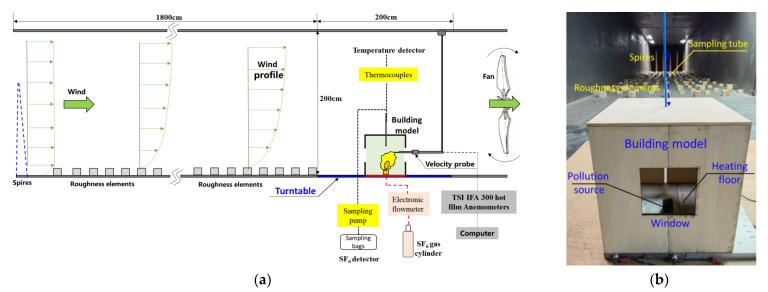
Wind-tunnel experimental model and setups. (**a**) Wind tunnel experimental setups. (**b**) Building model.

**Figure 5 ijerph-19-16889-f005:**
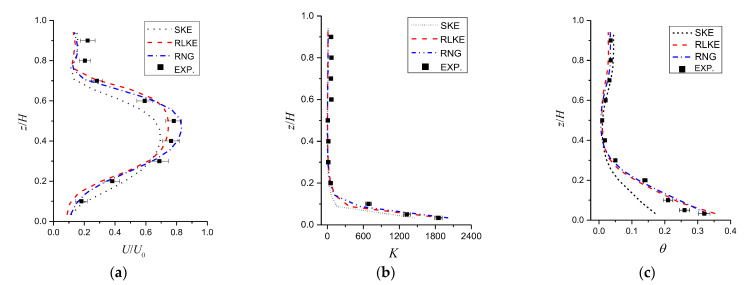
Model validation (standard, realizable, and RNG *k*-*ε* models vs. WT data) for cross-ventilation (*Ri* = 0.27). (**a**) *U/U*_0_, (**b**) *K* and (**c**) *θ* along the indoor vertical center line.

**Figure 6 ijerph-19-16889-f006:**
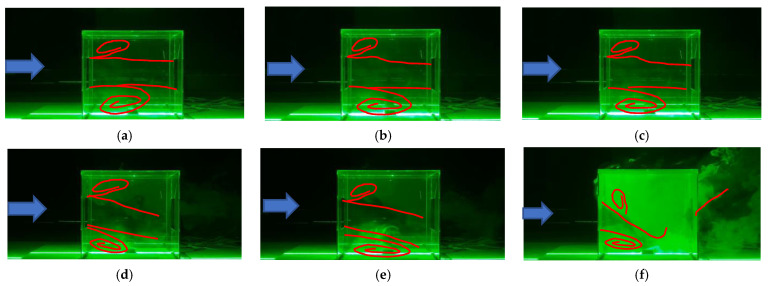
Flow and dispersion measured using the laser sheet-CCD concentration testing system under different *Ri*. (**a**) *Ri* = 0; (**b**) *Ri* = 0.04; (**c**) *Ri* = 0.27; (**d**) *Ri* = 1.86; (**e**) *Ri* = 5.31; (**f**) *Ri* = 11.11.

**Figure 7 ijerph-19-16889-f007:**
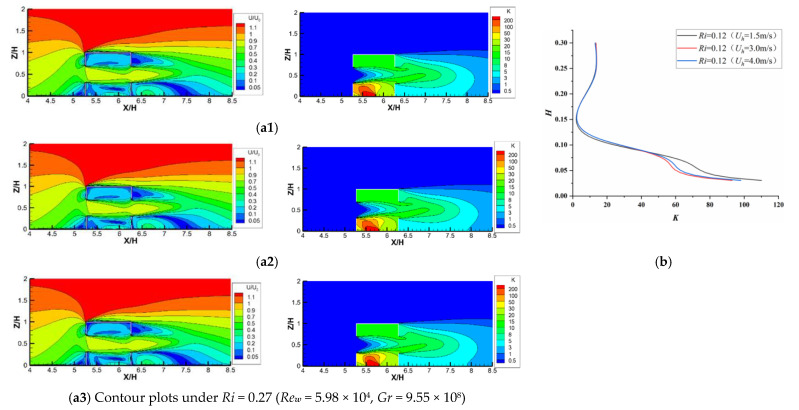
Comparison of *U/U_0_* and *K* values under the same *Ri* (= 0.27) with different *Re* for cross-ventilation. Contour plots under (**a1**) *Re**_w_*** = 2.26 × 10^4^, *Gr* = 1.36 × 10^8^; (**a2**) *Re*w = 4.50 × 10^4^, *Gr* = 5.61 × 10^8^); (**a3**) *Re**_w_*** = 5.98 × 10^4^, *Gr* = 9.55 × 10^8^; (**b**) *K* distributions along line A.

**Figure 8 ijerph-19-16889-f008:**
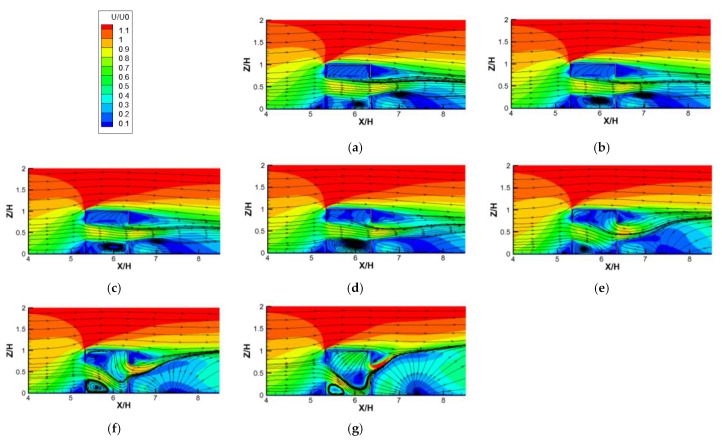
Contours of dimensionless velocity (*U*/*U_0_*) against increasing *Ri* numbers for cross-ventilation cases. (**a**) *Ri* = 0; (**b**) *Ri* = 0.04; (**c**) *Ri* = 0.27; (**d**) *Ri* = 1.86; (**e**) *Ri* = 5.31; (**f**) *Ri* = 11.11; (**g**) *Ri* = 26.65.

**Figure 9 ijerph-19-16889-f009:**
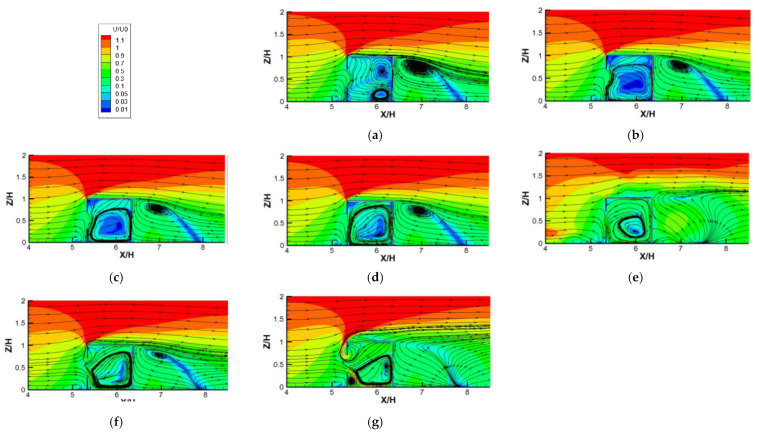
Contours of dimensionless velocity (*U*/*U_0_*) against increasing *Ri* numbers for single-sided ventilation cases. (**a**) *Ri* = 0; (**b**) *Ri* = 0.04; (**c**) *Ri* = 0.27; (**d**) *Ri* = 1.86; (**e**) *Ri* = 5.31; (**f**) *Ri* = 11.11; (**g**) *Ri* = 26.65.

**Figure 10 ijerph-19-16889-f010:**
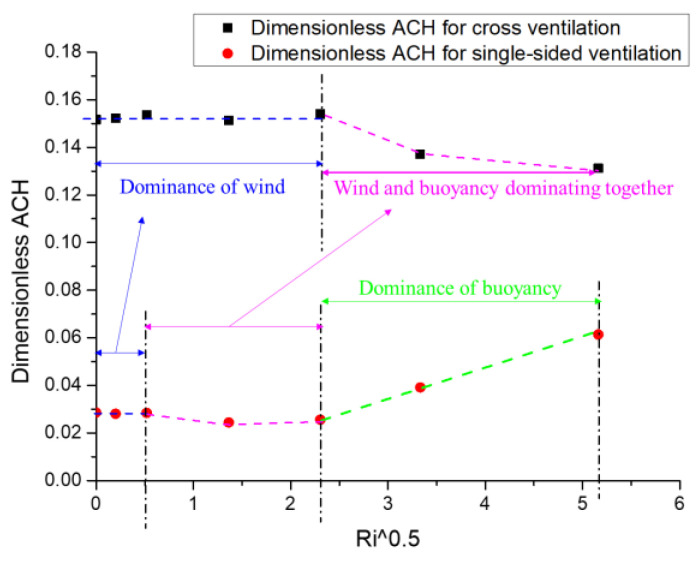
Dimensionless *ACH* against the square *Ri* numbers for cross- and single-sided ventilation (dotted lines in different color represent the *Ri* ranges of dominance of wind or buoyancy).

**Figure 11 ijerph-19-16889-f011:**
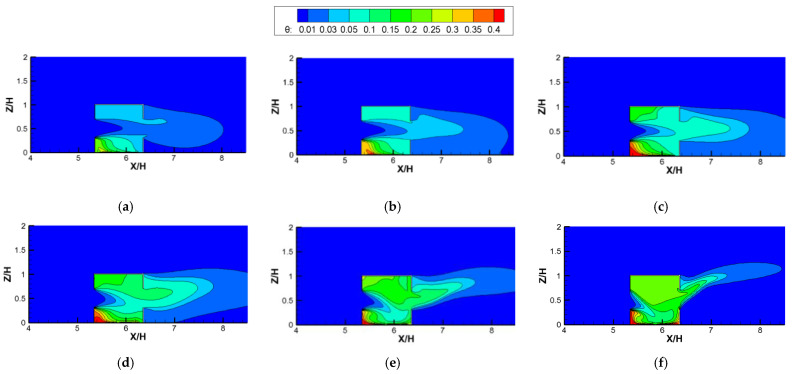
Contours of dimensionless temperature (*θ*) against increasing *Ri* numbers for cross-ventilation cases. (**a**) *Ri* = 0.04; (**b**) *Ri* = 0.27; (**c**) *Ri* = 1.86; (**d**) *Ri* = 5.31; (**e**) *Ri* = 11.11; (**f**) *Ri* = 26.65.

**Figure 12 ijerph-19-16889-f012:**
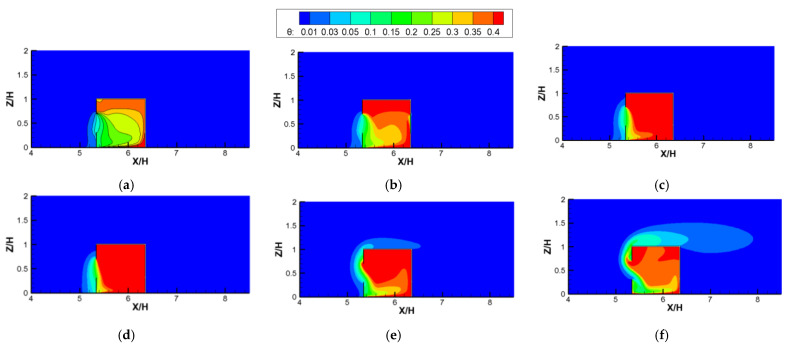
Contours of dimensionless temperature (*θ*) against increasing *Ri* values for single-sided ventilation cases. (**a**) *Ri* = 0.04; (**b**) *Ri* = 0.27; (**c**) *Ri* = 1.86; (**d**) *Ri* = 5.31; (**e**) *Ri* = 11.11; (**f**) *Ri* = 26.65.

**Figure 13 ijerph-19-16889-f013:**
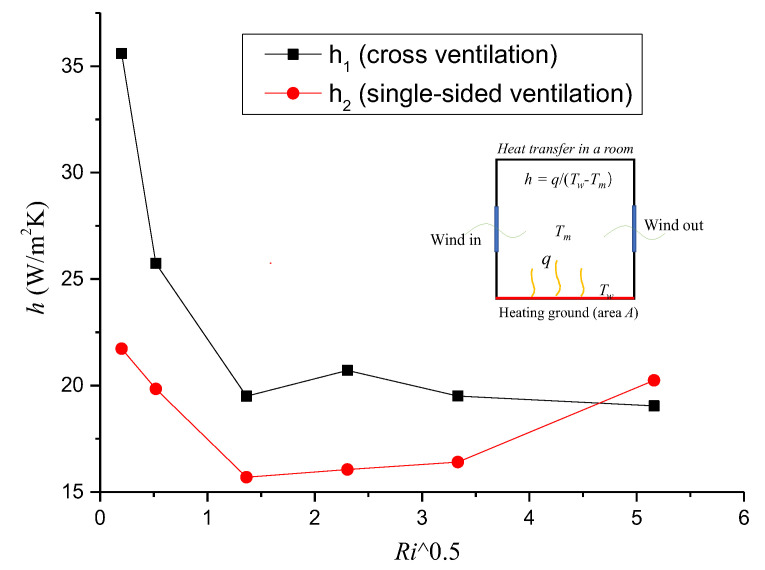
Mean *h* plotted against the square *Ri* numbers for cross- and single-sided ventilation cases.

**Figure 14 ijerph-19-16889-f014:**
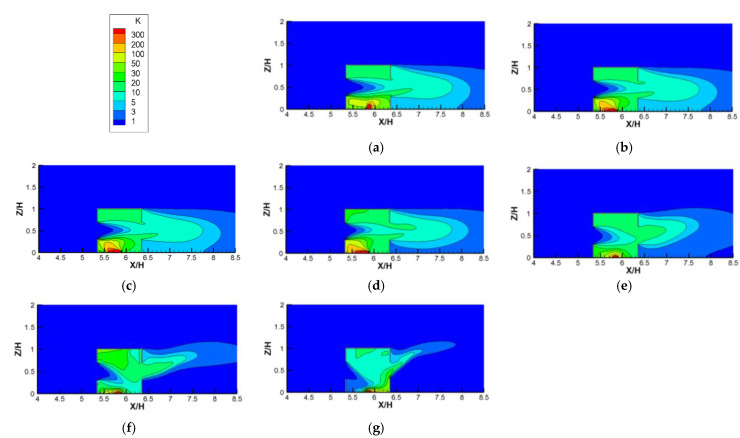
Contours of the dimensionless concentration (K) against increasing *Ri* values for cross-ventilation cases. (**a**) *Ri* = 0; (**b**) *Ri* = 0.04; (**c**) *Ri* = 0.27; (**d**) *Ri* = 1.86; (**e**) *Ri* = 5.31; (**f**) *Ri* = 11.11; (**g**) *Ri* = 26.65.

**Figure 15 ijerph-19-16889-f015:**
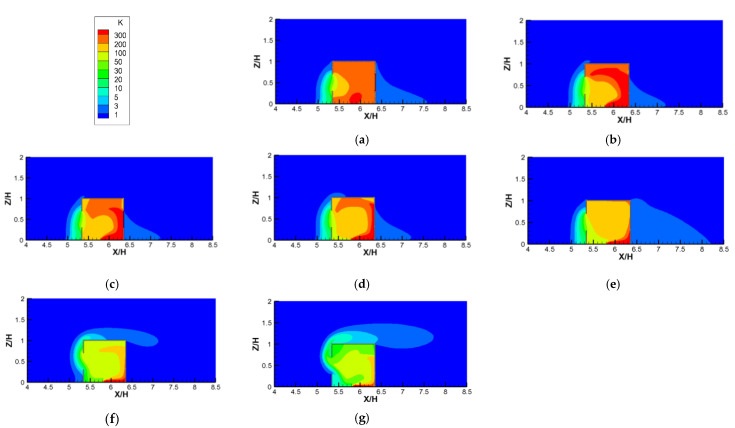
Contours of the dimensionless concentration (K) against increasing *Ri* values for single-sided ventilation cases. (**a**) *Ri* = 0; (**b**) *Ri* = 0.04; (**c**) *Ri* = 0.27; (**d**) *Ri* = 1.86; (**e**) *Ri* = 5.31; (**f**) *Ri* = 11.11; (**g**) *Ri* = 26.65.

**Figure 16 ijerph-19-16889-f016:**
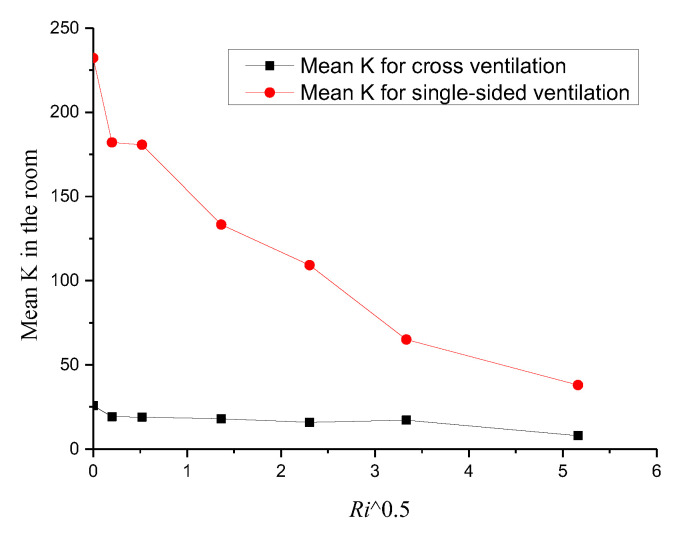
Mean *K* in the room, plotted against the square *Ri* numbers for cross- and single-sided ventilation.

**Table 1 ijerph-19-16889-t001:** The *Ri* cases studied under cross-ventilation conditions for reduced-scale physical models.

Cases	*U_H_*	*Re_H_*	*U_w_*	*Re_w_*	Δ*T* (K)	*Gr*	*Ri* = *Gr*/*Re_w_*^2^
1	6	1.36 × 10^5^	3.96	8.94 × 10^4^	0	0	0
2	6	1.36 × 10^5^	3.96	8.94 × 10^4^	60	3.08 × 10^8^	0.04
3 (a)	4	9.04 × 10^4^	2.65	5.98 × 10^4^	240	9.55 × 10^8^	0.27
3 (b)	3	6.78 × 10^4^	1.99	4.50 × 10^4^	120	5.61 × 10^8^	0.27
3 (c)	1.5	3.39 × 10^4^	1.00	2.26 × 10^4^	25	1.36 × 10^8^	0.27
4	1.5	3.39 × 10^4^	0.90	2.04 × 10^4^	180	7.74 × 10^8^	1.86
5	1	2.26 × 10^4^	0.59	1.34 × 10^4^	240	9.55 × 10^8^	5.31
6	0.8	1.81 × 10^4^	0.44	1.00 × 10^4^	300	1.11 × 10^9^	11.11
7	0.6	1.36 × 10^4^	0.30	6.84 × 10^3^	360	1.25 × 10^9^	26.65

**Table 2 ijerph-19-16889-t002:** The boundary conditions for the numerical models.

Boundary Conditions	Velocity	*k*	*ε*	Temperature	Pollutants
Inlet plane	$\frac{U_{z}}{U_{H}}={\left(\frac{z}{H}\right)}^{\alpha }$	$\frac{3}{2}{\left(u\left(z\right)I\left(z\right)\right)}^{2}$	Cμ34k32l	T_a_	0
Outlet plane	∂u∂x=0; *v* = *w* = 0	∂k∂x=0	∂ε∂x=0	∂T∂x=0	∂c∂x=0
Central plane	∂u∂y=0;∂w∂y=0; *v* = 0	∂k∂y=0	∂ε∂y=0	∂T∂y=0	∂c∂y=0
Top plane	∂u∂z=0;∂v∂z=0; w = 0	∂k∂z=0	∂ε∂z=0	∂T∂z=0	∂c∂z=0
Right/left plane	∂u∂y=0;∂w∂y=0; *v* = 0	∂k∂y=0	∂ε∂y=0	∂T∂y=0	∂c∂y=0
Building walls (Wood)	SWFs	0	0	Mixed	Zero gradient
Outdoor bottom plane	SWFs	0	0	*h* = 0	Zero gradient
Indoor ground (Aluminum)	SWFs	0	0	*T_f_*	Zero gradient
Pollutant source	0	0	0	/	0.025 g/(m^3^·s)

**Table 3 ijerph-19-16889-t003:** Dimensionless air exchange rate (*ACH*+/−) for cross-ventilation under different *Ri* cases.

Cases	*Ri*	$\bar{ACH}+/\left(V/T\right)$	$ACH^{'}+/\left(V/T\right)$	$\bar{ACH}-/\left(V/T\right)$	$ACH^{'}-/\left(V/T\right)$	$ACH+/\left(V/T\right)\;$	$ACH-/\left(V/T\right)\;$	$\bar{ACH}+/ACH+$ $ACH^{'}+/ACH+$	$\bar{ACH}-/ACH-$ $ACH^{'}-/ACH-$
1	0	0.1049	0.0433	0.1118	0.0433	0.1482	0.1551	0.71/0.29	0.72/0.28
2	0.04	0.1059	0.0441	0.1102	0.0444	0.1500	0.1546	0.71/0.29	0.71/0.29
3	0.27	0.1064	0.0443	0.1121	0.0443	0.1507	0.1564	0.71/0.29	0.72/0.28
4	1.86	0.1035	0.0439	0.1114	0.0439	0.1474	0.1553	0.70/0.30	0.72/0.28
5	5.31	0.0948	0.0532	0.1067	0.0532	0.1480	0.1599	0.64/0.36	0.67/0.33
6	11.11	0.0889	0.0408	0.1037	0.0408	0.1297	0.1445	0.69/0.31	0.72/0.28
7	26.65	0.0770	0.0444	0.0968	0.0444	0.1214	0.1411	0.63/0.37	0.69/0.31

**Table 4 ijerph-19-16889-t004:** Dimensionless air exchange rate (*ACH*+/−) for single-sided ventilation under different *Ri* cases.

Cases	*Ri*	$\bar{ACH}+/\left(V/T\right)$	$ACH^{'}+/\left(V/T\right)$	$\bar{ACH}-/\left(V/T\right)$	$ACH^{'}-/\left(V/T\right)$	$ACH+/\left(V/T\right)\;$	$ACH-/\left(V/T\right)\;$	$\bar{ACH}+/ACH+$ $ACH^{'}+/ACH+$	$\bar{ACH}-/ACH-$ $ACH^{'}-/ACH-$
1	0	0.0069	0.0216	0.0071	0.0216	0.0285	0.0287	0.24/0.76	0.25/0.75
2	0.04	0.0069	0.0215	0.0062	0.0215	0.0284	0.0278	0.24/0.76	0.22/0.78
3	0.27	0.0063	0.0220	0.0066	0.0220	0.0283	0.0286	0.22/0.78	0.23/0.77
4	1.86	0.0028	0.0219	0.0023	0.0219	0.0247	0.0242	0.11/0.89	0.09/0.91
5	5.31	0.0043	0.0214	0.0040	0.0214	0.0258	0.0254	0.17/0.83	0.16/0.84
6	11.11	0.0188	0.0175	0.0244	0.0175	0.0364	0.0420	0.52/0.48	0.58/0.42
7	26.65	0.0354	0.0182	0.0510	0.0182	0.0536	0.0692	0.66/0.34	0.74/0.26

## Data Availability

Not applicable.

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
