# Peer review of "Effects of Radiant Floor Heating Integrated with Natural Ventilation on Flow and Dispersion in a Newly Decorated Residence"

_ijerph, 2022, doi:10.3390/ijerph192416889_

Round 1
Reviewer 1 Report
In this study, a three-dimensional Computational Fluid Dynamic (CFD) model coupled with RNG turbulence model was adopted to try to investigate the effects of radiant floor heating integrated with natural ventilation on indoor flow and pollutant dispersion by defining the Ri numbers from 0 to 26.65. It can be seen that the author has done a very meticulous work, and the research conclusions are also quite valuable. Before publication, I hope the comments below can be helpful for the authors.
(1) The use of wind tunnel experiments to simulate the thermal environment is a complicated problem, especially the accuracy of the model is often questioned, the author should cite relevant references in Introduction to support the research method of this paper.
(2) The introduction is too tedious, so the author can appropriately delete the irrelevant literature review to highlight the importance of the research in this paper.
(3) Some variables and subscripts should be revised according to the journal requirements.
(4) Reconsider 2.2.2 mesh generation, the author should applied XXH to replace the real mesh resolution, such as 2-5mm. And this section should be further reduced.
(5) Table 2 lacks the boundary conditions of the left plane, and the boundary conditions of the pollutants on the wall should not be 0, but should be zero gradient. In addition, boundary conditions have been listed in Table 2, Eqs. 9 and 10 should be omitted.
(6) Reconsider Lines 243-269, because in Table 2 boundary conditions have been listed.
(7) Recheck the full text because there are some grammatical and spelling errors.
(8) Reconsider the conclusion (1), model validation should not be taken as a conclusion. In addition, the conclusion is too long and should be further simplified and condensed.
Author Response
Answers to the reviewers’ comments
Reviewer #1
General comments:
In this study, a three-dimensional Computational Fluid Dynamic (CFD) model coupled with RNG turbulence model was adopted to try to investigate the effects of radiant floor heating integrated with natural ventilation on indoor flow and pollutant dispersion by defining the Ri numbers from 0 to 26.65. It can be seen that the author has done a very meticulous work, and the research conclusions are also quite valuable. Before publication, I hope the comments below can be helpful for the authors.
Answer: thanks for the reviewer’s affirmation.
- The use of wind tunnel experiments to simulate the thermal environment is a complicated problem, especially the accuracy of the model is often questioned, the author should cite relevant references in Introduction to support the research method of this paper.
Answer: accepted. Reference [35] has been added. Zhou et al. [35] conducted the experimental and numerical studies to investigate the effects of ventilation and flow heating systems on the dispersion and deposition of fine particles in an enclosed environment. Their study would be helpful for the design of ventilation and heating system to remove PM pollution in the room.
35、Y. Zhou, Y. Deng, P, Wu, S. Cao, The effects of ventilation and floor heating systems on the dispersion and deposition of fine particles in an enclosed environment, Building and Environment, 125(2017) 192-205.
- The introduction is too tedious, so the author can appropriately delete the irrelevant literature review to highlight the importance of the research in this paper.
Answer: accepted. The introduction has been shorten, please see the revised version.
- Some variables and subscripts should be revised according to the journal requirements.
Answer: accepted. All the variables and subscripts have been modified.
- Reconsider 2.2.2 mesh generation, the author should applied XXH to replace the real mesh resolution, such as 2-5mm. And this section should be further reduced.
Answer: accepted. this part has been revised as: The computational domain was discretized by ANSYS ICEM 18.0 by the structured grids with variable sizes: the finest grid resolution with the cell magnitude of 0.02–0.05H inside the indoor scale, a coarser resolution close to the building and ground surfaces with the magnitude of 0.05–0.1H, and a gradually coarse mesh from the building and ground surfaces to the domain boundaries by using the increasing ratio of 1.12–1.16.
- Table 2 lacks the boundary conditions of the left plane, and the boundary conditions of the pollutants on the wall should not be 0, but should be zero gradient. In addition, boundary conditions have been listed in Table 2, Eqs. 9 and 10 should be omitted.
Answer: accepted. Thanks for the revision. Table 2 has been revised.
- Reconsider Lines 243-269, because in Table 2 boundary conditions have been listed.
Answer: accepted, thanks for the suggestion. this part in Lines 243-269 has been removed.
- Recheck the full text because there are some grammatical and spelling errors.
Answer: addressed, thanks for the suggestion. The grammatical and spellings errors have been modified by the Languish editing system: https://www.mdpi.com/authors/english.
- Reconsider the conclusion (1), model validation should not be taken as a conclusion. In addition, the conclusion is too long and should be further simplified and condensed.
Answer: addressed, thanks for the suggestion. Conclusion (1) has been removed and the other conclusions have also been simplified. Please see the revised version.

Reviewer 2 Report
-What scientific gaps this study aims to fill?
-Please clarify why indoor radiant heating floor is integrated with natural ventilation. Usually, when radiant cooling or heating is running, the window is closed to save energy.
-There is no need to mention COVID-19.
-How about the heating capacity/loss under different natural ventilation modes?
-It is weird to see that the room has no fresh air system but an open window instead.
-Have the authors considered indoor thermal comfort when opening the window?
-The English language should be improved.
Author Response
Answers to the reviewers’ comments
Reviewer #2
(1)What scientific gaps this study aims to fill?
Answer: addressed, thanks for the suggestion. The aim of this study has been added in the end of Introduction, as follows: The results of this study may be helpful in promoting the rapid release and removal of pollutants from building materials in newly decorated rooms through ventilation and heating systems.
(2)Please clarify why indoor radiant heating floor is integrated with natural ventilation. Usually, when radiant cooling or heating is running, the window is closed to save energy.
Answer: addressed, thanks for the suggestion. The scenario simulated in this study is that in the newly decorated and unoccupied residence, the heating system can promote the release of VOCs such as formaldehyde from the building materials, and then the pollutants can be quickly removed from the room by the window ventilation. Therefore, the relative description about the case design has been modified in the revised version in the beginning of the last paragraph of Introduction, as follows, “Usually, in a newly decorated and unoccupied residence, heating process can promote the release of toxic and harmful substances such as formaldehyde from building materials, and natural ventilation can effectively and quickly discharge these indoor pollutants. Even when the occupied residences are heated in winter, a fresh air system or reasonable window ventilation is needed to deliver fresh air indoors.”
Then to make it clearer to the readers and reviewers, the title of this study also has been modified as follows: Effects of radiant floor heating integrated with natural ventilation on flow and dispersion in a newly decorated residence.
(3)There is no need to mention COVID-19.
Answer: addressed. Thanks for this suggestion. This part about COVID-19 has been removed, and the Introduction has been modified, please see the revised version.
(4)How about the heating capacity/loss under different natural ventilation modes?
Answer: Thanks for this comment. This paper discussed the mean convective heat transfer coefficient h (Section 4.4.2) to evaluate indoor and outdoor heat exchange under different Ri. In addition, the purpose of this study is to promote the rapid release and removal of the pollutants from building materials in newly decorated rooms by window ventilation. Therefore, the limitations of this study have been added in the end of this paper, as follows:
Limitations: The scenario simulated in this study was that in a newly decorated and unoccupied residence, in which the heating system could promote the release of VOCs such as formaldehyde from the building materials, and then natural ventilation from windows could quickly remove the pollutants from the room, so there was no discussion of thermal comfort. In future works, we will continue to investigate the coupled effects of indoor heating and fresh air systems on indoor pollutant removal and thermal comfort in occupied residences.
(5)It is weird to see that the room has no fresh air system but an open window instead.
Answer: Thanks for this comment. In a newly decorated and unoccupied residence, the pollutants released from the building materials can be effectively and economically by open window ventilation.
(6)Have the authors considered indoor thermal comfort when opening the window?
Answer: Thanks for this comment. This paper did not discuss thermal comfort, but discussed the dimensionless temperature distributions (Section 4.4.1) under different Ri. In addition, the purpose of this study is to promote the rapid release and removal of the pollutants from building materials in newly decorated rooms by window ventilation. Therefore, the limitations of this study have been added in the end of this paper, as follows:
Limitations: The scenario simulated in this study was that in a newly decorated and unoccupied residence, in which the heating system could promote the release of VOCs such as formaldehyde from the building materials, and then natural ventilation from windows could quickly remove the pollutants from the room, so there was no discussion of thermal comfort. In future works, we will continue to investigate the coupled effects of indoor heating and fresh air systems on indoor pollutant removal and thermal comfort in occupied residences.
(7)The English language should be improved.
Answer: addressed, thanks for the suggestion. The grammatical and spellings errors have been modified by the Languish editing system: https://www.mdpi.com/authors/english. Please see the attachment

Round 2
Reviewer 2 Report
Most of the issues have been well addressed.